# Peptoids: Smart and Emerging Candidates for the Diagnosis of Cancer, Neurological and Autoimmune Disorders

**DOI:** 10.3390/ijms242216333

**Published:** 2023-11-15

**Authors:** Anna Giorgio, Annarita Del Gatto, Simone Pennacchio, Michele Saviano, Laura Zaccaro

**Affiliations:** 1Department of Chemical Sciences, University of Padova, 35131 Padova, Italy; anna.giorgio@phd.unipd.it; 2Institute of Biostructure and Bioimaging (IBB), CNR, 80131 Naples, Italy; annarita.delgatto@cnr.it; 3Interuniversity Research Centre on Bioactive Peptides (CIRPeB) “Carlo Pedone”, University of Naples “Federico II”, 80131 Naples, Italy; 4Institute of Condensed Matter Chemistry and Technologies for Energy (ICMATE), CNR, 35127 Padova, Italy; simonepennacchio94@gmail.com; 5Institute of Crystallography (IC), CNR, 70126 Bari, Italy; michele.saviano@cnr.it

**Keywords:** peptoids, diagnosis, targeted imaging, blood-based screening, cancer, neurological disease, autoimmune disease

## Abstract

Early detection of fatal and disabling diseases such as cancer, neurological and autoimmune dysfunctions is still desirable yet challenging to improve quality of life and longevity. Peptoids (N-substituted glycine oligomers) are a relatively new class of peptidomimetics, being highly versatile and capable of mimicking the architectures and the activities of the peptides but with a marked resistance to proteases and a propensity to cross the cellular membranes over the peptides themselves. For these properties, they have gained an ever greater interest in applications in bioengineering and biomedical fields. In particular, the present manuscript is to our knowledge the only review focused on peptoids for diagnostic applications and covers the last decade’s literature regarding peptoids as tools for early diagnosis of pathologies with a great impact on human health and social behavior. The review indeed provides insights into the peptoid employment in targeted cancer imaging and blood-based screening of neurological and autoimmune diseases, and it aims to attract the scientific community’s attention to continuing and sustaining the investigation of these peptidomimetics in the diagnosis field considering their promising peculiarities.

## 1. Introduction

The early and accurate diagnosis of deadly and disabling diseases such as cancers and neurological syndromes is constantly a challenge for the research community. The identification of early biomarkers and the development of related molecular probes are crucial to classify those patients who could best benefit from a specified therapeutic regimen, dose, or duration of the drug prolonging their survival and quality of life. Recently, different methodologies for the identification of specific, smart, and selective probes have been proposed. The design of peptide-based systems has been one of the most successful approaches for the discovery of new probes. Thanks to the advantages of using short synthetic peptides over recombinant proteins and small molecules [1], such as high selectivity, specificity, non-immunogenicity, low cost of production, and high degree of chemical diversity, all features tightly correlated to peptide architectures. Unfortunately, their acceptance in diagnostics has very often been hindered by the low resistance to proteolytic degradation. Peptidomimetics are a worthy alternative to peptides for circumventing such complications, and some of them have entered clinical trials as drugs [2,3]. Peptoids (N-substituted glycine oligomers) represent one of the most promising classes of peptidomimetics, with a backbone like a polyglycine, featured with the side-chains covalently linked to the amide nitrogen on the peptoid.

Zuckermann and coworkers in 1992 [4], published the invention of the sub-monomer solid-phase synthesis method for peptoids preparation (Figure 1). This strategy has greatly increased the chemical diversity, the synthetic efficiency and yield, reducing time and costs.

The shift of the side chain on the amide nitrogen improves the enzyme resistance (Figure 2) and their chemical and thermal stability. The enhanced stability of peptoids is reasoned to prevent premature drug activity caused via enzymatic degradation in vivo.

In the last decade, many research studies have demonstrated different applications of peptoids in biomedicine and material science [5]. Some of them act as molecular modulators, e.g., ligands for GPCRs [6] and protease inhibitors [7], novel bio-inspired nanomaterials [8,9], potent antiviral [10] and anti-cancer agents [11], and notably, antimicrobials [12,13,14,15,16].

Nowadays, pathology diagnosis relies mainly on targeted molecular imaging techniques, such as nuclear medicine modalities PET and SPECT, which use radioligands and provide detailed metabolic and functional information and accurate data, thus revealing disease progression. PET and SPECT imaging are mainly employed in oncology and neurology, and are characterized by a good sensitivity, spatial resolution, and unlimited penetration depth, leading to their wide use for preclinical and clinical studies. The use of specific contrast agents, such as peptides and peptoids for an array of disease-related receptors or surrogate biomarkers, has contributed to improving the targeting efficiency of the detection by eliminating misinterpretation of false-positive outcomes. Nevertheless, PET and SPECT imaging applications are entirely dependent on the availability of radiotracers whose development is the result of different steps. These steps include selection of suitable biological targets, design and identification of promising targeting molecules, optimization of the radiolabeling strategies, in vitro and in vivo preclinical assessment of candidate radiotracers [17].

MRI is a noninvasive medical imaging technique extensively used for anatomical imaging of soft body-tissues [18]. MRI is traditionally regarded as a safe imaging modality due to the absence of ionizing radiations. MR contrast agents are usually administered in high doses and consequently are generally considered too insensitive for molecular imaging applications. As a result, novel MR contrast agents based on nanoparticle or dendrimer platforms have been developed to target specific biostructures, significantly amplifying the additive effect of multiple paramagnetic centers over a single center.

Neurological and autoimmune diseases are common dysfunctions that affect millions of people [19,20]. Some autoimmune diseases are often difficult to diagnose because their early signs and symptoms are not specific and can look like signs and symptoms of other illnesses. In neurological dysfunctions, an early diagnosis frequently comes far too late, and the treatment after the onset of the pathology can make it hard to stop or reverse the evolution of the disease. In Alzheimer’s disease (AD) patients, who are usually asymptomatic in the early stages, a timely diagnosis is even more difficult. Most neurological dysfunction exams and texts occur via imaging, blood markers, and cerebrospinal fluid detection. The latter is a reliable and accurate method, but its invasiveness limits its application in diagnosis and pathology progression monitoring. As an alternative, blood tests are inexpensive, minimally invasive, and very promising for the identification of biomarkers in the earliest and asymptomatic stages of some neurological and autoimmune diseases and in large-scale screening.

To our knowledge, the current review is the only one reported in the last decade in the field of peptoids in the diagnosis and plays a role of great importance to the life science and chemistry community. It well fits into this context and summarizes recent advancements regarding the promising applications of peptoid oligomers as new and versatile platforms for cancer targeted imaging using PET, MRI and blood-based screening in the diagnosis of neurological and autoimmune diseases for an effective and early detection of these pathologies with great impact on the individuals, families, and society.

## 2. Radiolabeled Peptoids and Peptoid/Peptide Hybrids for Cancer-Targeted Imaging

The success of an accurate and reliable diagnosis is correlated with the identification of biomarkers associated with a given pathology. The detection of well-researched and widely accepted biomarkers using tailored probes is fundamental for effective imaging targeted approaches, reducing the frequency of false positive diagnoses and bringing down overall healthcare costs.

To illustrate the advances made in cancer-targeted imaging, the peptoid oligomers and peptide/peptoid hybrids known to date are shown below (Figure 3). These oligomers are useful for the targeted imaging of different biomarkers, such as VEGFR2 and NTS1/NTS2.

### 2.1. VEGFR2 Biomarker

Several angiogenesis-related biomarkers have been discovered and used as imaging targets so far. Vascular endothelial growth factors receptor 2 (VEGFR2) plays a fundamental role in promoting tumor angiogenesis and tumor growth. Recently, a VEGFR2 binding peptoid sequence GU40C (9 mer) was selected via a large library screening [21], but the molecule shows a low binding affinity for the target limiting its development as receptor-based imaging reagent. The same peptoid synthesized as dimer (GU40C4) was developed and proven to have a high affinity against the VEGFR2 activation and an effect in inhibiting angiogenesis and tumor growth in vivo [21]. In 2010, De Leòn-Rodrìgues et al. [22] created targeted agents for MRI made up of GU40C4 and a poly(DOTA-lysine) dendron (Gd_8_ dendron) with a lysine linker (Table 1). A Cys residue was added to the C-terminus of the peptoid to maintain the conjugation site away from the central pharmacophore and allow a site-specific coupling with DOTA through the maleimide-thiol coupling chemistry. DOTA was chosen as Gd^3+^ chelating agent because it forms complexes with a high level of thermodynamic and kinetic stability, eliminating issues with the metal release, especially for in vivo use. Two conjugates with different linkers between GU40C4 and Gd_8_-dendron were investigated, and the best of them (conjugate 2 reported in De León-Rodríguez et al. [22], Table 1) was selected for the binding affinity to the receptor and the molecular r1 relaxivity value. Considering the interesting MR imaging results obtained on porcine aortic endothelial cells expressing VEGFR2, the MRI properties of the compound were also evaluated in vivo by using MDA-MB-231 tumor xenografts known to express VEGFR2 [23]. Tumor uptake in mice treated with the peptoid-Gd_8_ dendron was maximally enhanced at ∼4 h post-injection, while the image intensity after injection of a scrambled peptoid Gd_8_-dendron conjugate reverted to baseline levels by 4 h. The kidneys intensity indicated the compound does not accumulate in these organs until much later in the study, but it is localized elsewhere during the early time points. At 48–72 h after injection, the kidney signal intensity in mice treated with the specific and the scrambled peptoid was restored to pre-treatment levels. All the results showed that this platform can give a promising T1 probe for MRI imaging of VEGFR2.

In 2011, Hao et al. [24] evaluated the potential of the dimeric peptoid GU40C4 labeled with positron emitter ^64^Cu to be used for tumor visualization in a prostate cancer mouse model highly expressing VEGFR2, exploiting the relatively small size of the molecule, the high in vivo stability and binding affinity to the receptor. The tumor uptake level of ^64^Cu-DOTA-GU40C4 was relatively low when compared to other reported VEGFR2-targeted imaging agents. This is likely due to different tumor models expressing dissimilar levels of VEGFR2 and non-specific tumor uptake effects which can occur. Nevertheless, the steady tumor uptake retention and effective clearance from non-target organs within the 20 h period of ^64^Cu-labeled homodimer allowed for superior tumor imaging contrast. This can be attributed to the peptoid cell permeability, provided by its positive charge [25] and the rapid in vivo kinetics which are correlated with its low molecular weight. Unfortunately, as the most reported PET probes for VEGFR2 imaging, GU40C4 binds to both VEGFR1 and VEGFR2 with a similar affinity [26] and thus, the discovery of novel peptoids with higher binding selectivity is required for this purpose.

### 2.2. NTS1/NTS2 Biomarker

The neurotensin receptor 1 (NTSR1) is an attractive target for the development of diagnostic and therapeutic drugs since its role in the onset and progression of many tumors such as breast, prostate, and colorectal [27] is well documented [28].

The neurotensin peptide (NT) shows a short half-life in blood and is not suitable for direct radiolabeling and application [29,30]. Accordingly, most advancements in neurotensin receptor imaging have been addressed at enhancing NT serum stability and NTSR1 specificity [31].

Maschauer et al. [32] reported a successful strategy to produce ^18^F-glycopeptoid derivatives of NT as PET imaging agents, combining the ^18^F-labeling with the glycosylation required for the improvement of the biokinetic and in vivo clearance properties. In detail, they synthesized some glycopeptide/peptoid hybrid analogs of NT 8–13, which is the highest active C-terminal hexapeptide of the endogenous NT, by using the CuACC click chemistry for the introduction of the glycosyl moiety at the N-terminus of the peptide. According to previous studies on the effects of the ligand conformation and peptide backbone modifications on affinity changes for several NT 8–13 analogs [33], metabolic stabilization was predicted to be achieved by the replacement of the N-terminal Arg-Arg by the peptoid-like residue N-(4-aminobutyl) Gly-Lys (NLys-Lys). The stabilized derivative [^18^F]FGlc-NT4 (Table 1) showed specific binding in HT29 cells expressing a high level of NTSR. Rat brain slices used for in vitro autoradiography further evidenced the [^18^F]FGlc-NT4 selective binding to NTSR-rich regions, which were fully blocked in the presence of NT. The biokinetics behavior of [^18^F]FGlc-NT4 in HT29 xenografted nude mice was also evaluated assessing metabolic stability in vivo and an appropriate signal-to-noise ratio for PET imaging at the earliest time points after injection. Although the results obtained are promising, further studies will focus on improving kidney clearance by changing the glycosyl moiety of the ^18^F-glycopeptoids to promote their use as tracers.

The same authors also investigated the properties of the peptide/peptoid hybrid NT4, radiolabeled with ^68^Ga for the visualization of NTSR1-expressing tumors by PET [34] since one of the major obstacles to using ^18^F is the high-cost production of the radionuclide by the cyclotron; the short-lived ^68^Ga is provided by the ^68^Ge/^68^Ga generator, which allows frequent and easy access to the isotope. Because elongation of the N-terminus of NT 8–13 is usually well-tolerated with respect to NTSR recognition, the DOTA chelator was inserted at the N-terminal ending. NT receptor binding studies using hNTS1-expressing CHO cells indicated that the Ga-DOTA, conjugate [^68^Ga]3 (Table 1) showed a lower affinity in comparison with NT 8-13 and the mimetic [^18^F]FGlc-NT4. Nevertheless, the lower affinity was balanced by the ability of the molecule to internalize into tumor cells via NTSR1, a key factor for in vivo tumor visualization. Biodistribution studies in HT29 xenografted nude mice revealed metabolic stability in vivo, fast blood clearance, high uptake in kidneys, and low uptake in the liver. Most importantly, the uptake of ^68^Ga-3 in the HT29 tumor was comparable to the ^18^F-labeled glycopeptide/peptoid hybrid, and the kidney uptake and clearance was improved respect to [^18^F]FGlc-NT4. The in vivo specificity and affinity of NTSR1-mediated uptake of [^68^Ga]3 was demonstrated in animals co-injected with [^68^Ga]3 and NT4.

A few publications have shown a correlation between tumor progression and expression of the NTSR2 subtype [35] indicating this receptor is also an interesting biomarker for personalized (precision) medicine of cancer. In 2015, Maschauer et al. [36] focused on the synthesis and in vivo evaluation of ^18^F4 (Table 1). This is a peptide/peptoid hybrid derived from NT 8-13 that was identified by the selection of a series of hybrids [37,38,39]. The molecule was significantly labeled with ^18^F isotope applying the strategy previously reported for [^18^F]FGlc-NT4 [32]. In vitro studies revealed the proteolytic stability of the derivative, and the binding affinity studies evidenced the outstanding subtype selectivity over NTSR1. In vitro autoradiography analyses of rat brain slices confirmed the ability of the molecule to discriminate between NTSR1 and NTSR2, differently distributed in the brain. Biodistribution studies using PC3 and HT29 xenografted nude mice showed successful retention of the tracer in the tumor, high accumulation in the kidneys, and moderate to low uptake in all other organs. Nevertheless, stability studies in vivo indicated a fast degradation of the compound in mouse blood in contrast to the high proteolytic resistance observed in vitro and further optimization of the compound is required.

**Table 1 ijms-24-16333-t001:** Radiolabeled peptoid and peptide/peptoid hybrid sequences for cancer-targeted imaging. The structure of each monomer is shown in the figure at the end of Section 3.5 and the respective amines used to synthesize them are shown in the table at the end of Section 3.5.

Name	Sequence	Ref.
Gd_8_-dendron-GU40C4	GU40C * -Ahx-bAla-Lys(GU40C *)-Cys[linker-(Gd_8_-dendron) **]-NH_2_	[22]
^64^Cu-DOTA-GU40C4	GU40C *-Ahx-bAla-Lys(GU40C *)-Cys(^64^Cu-maleimide-monoamide-DOTA derivative)-NH_2_	[24]
[^18^F]FGlc-NT4	Pra *** (2^18^FGlc)-*N*Lys-Lys-Pro-Tyr-Tle-Leu-OH	[32]
[^68^Ga]3	^68^Ga-DOTA-*N*Lys-Lys-Pro-Tyr-Tle-Leu-OH	[34]
^18^F-4	Pra *** (6^18^FGlc)-NMeArg-Arg-Pro-*N*homoTyr-Ile-Leu-OH	[36]

* GU40C = NLys-NLeu-NLys-Nmba-Npip-NLys-NLeu-NLys-NLys. ** (Gd8-dendron) = poly(GdDOTA)lysine dendrimer scaffold. *** Pra: propargyl Glycine to allow 18F-fluoroglycosylation via CuAAC click chemistry.

## 3. Peptoids for Blood-Based Screening of Neurological and Autoimmune Diseases

Fluid biomarkers and autoantibodies present in various biosamples are key hallmarks of severe pathologies and extremely useful for an early diagnosis. Non-invasive and label-free assays based on the search of these hallmarks in serum/plasma samples are deemed attractive prognostic approaches in the detection of neurological and autoimmune diseases. The wide application of blood-based screening depends on the development and application of ultrasensitive detection methods.

For this purpose, peptoid oligomers are widely applied in blood-based screening in the diagnosis of neurological and autoimmune diseases, such as Alzheimer’s disease (AD), Parkinson Disease (PD), Neuromyelitis Optica (NMO), Systemic Lupus Erythematosus (SLE), and Autism Spectrum Disorder (ASD) (Figure 4).

### 3.1. Alzheimer’s Disease

Reddy et al. [40] developed and validated a non-invasive assay for discovering IgG serum biomarkers in Alzheimer’s disease patients, an approach that does not need the knowledge of native antigens. They proved that microarrays displaying thousands of octameric peptoids, each bearing a Cys residue at the C-terminus for the covalent link to the array, can be used to simultaneously isolate IgG antibody biomarkers via the immunoassay method candidate. Selective peptoid ligands can pull antibodies out of the blood even if they are not able to bind them as well as the native antigens. This approach is rather different from previously reported screens of peptide libraries [41,42], which aim at the identification of a native epitope or a very close relative. The serum of six patients with AD, six control individuals, and six patients with PD were initially analyzed for the comparative screening of peptoid libraries and three molecules ADP1-3 (Table 2) that were able to capture IgG antibodies were successfully selected. Moreover, the screening of serum samples from 16 different patients with AD, 16 new controls, and samples from 6 subjects with SLE indicated that the three peptoids have a good sensitivity and specificity for the diagnosis of AD.

In 2015, Zhao and colleagues [43] merged the peptoid microarray with surface plasmon resonance imaging (SPRi) aiming to develop a faster and more routine diagnostic method for detecting AD biomarkers in serum. SPRi is a label-free and high-throughput biosensor technology broadly employed for investigating interactions between biomolecules occurring close to the SPR-active surface. The authors demonstrated that high concentrations of ADP3 immobilized on the chip can identify AD patients by binding to Aβ42, a crucial prognostic indicator generated by the sequential proteolysis of the amyloid precursor protein [44]. The estimation of the binding between ADP3 and Aβ42 through kinetic analysis indicated the high affinity of ADP3 to Aβ42 that can be ascribable to the ADP3 assembling into nanoclusters as well as the high concentration of peptoid on the surface.

In 2017, an antibody mimetic platform consisting of a self-assembled nanosheet peptoid displaying on both surfaces the ADP3 sequence was developed [45] (Table 2). In detail, the ADP3 peptoid was embedded in the middle of the amphiphilic peptoid sequence containing N-(2-carboxyethyl)glycine and N-(2-aminoethyl) glycine as the hydrophilic monomers and the N-(2-phenylethyl) glycine and N-(2-(4-biphenyl)ethyl) glycine as the hydrophobic monomers. The latter are able to self-assemble into a peptoid nanosheet thanks to the hydrophobic and electrostatic interactions between the monomers.

The nanosheets were featured using fluorescence microscopy and atomic force microscopy; the presence of the protrusion associated with the peptoid loop was thus confirmed. Initially, the ADP3 loop-bearing peptoid scaffold was immobilized on the sensor chip and used to calculate the binding affinity to Aβ42 by SPRi to test the potential of the self-assembled system to serve as a biosensor. Moreover, nanosheets with a different % to the loop were tested, indicating that the Aβ42 capture depends on the ADP3 loop concentration. In the presence of 100% loop content, the binding affinity is indeed higher than that of the linear-shaped ADP3 molecule itself and is likely due to the favorable accessibility of the loop on the nanosheet surface acting as a steady platform as well as to the increased density of the binding sites totally free for the molecular recognition and reduction of the nonspecific interactions. The authors demonstrated then the high efficiency of the nanosheet-based sensor to discriminate normal sera from AD patients and age-matched and non-demented control individuals with high sensitivity. Thus, unveiling the great potential of this antibody mimetic as a blood-based, early AD diagnosis.

Amnestic mild cognitive impairment (aMCI) is considered a precursor stage that can evolve into Alzheimer’s disease in which symptoms start 10–20 years before the onset of clinical evidence, making the timely diagnosis of the pathology difficult. Thus, the early detection of aMCI is a key opportunity to prevent the progression of dementia into AD. The peptoid ADP3 nanosheet as a label-free sensor was evaluated to distinguish the plasma and sera of patients with AD and aMCI from healthy individuals using SPRi [46,47]. Interestingly, the authors proved how this peptoid-based sensing system can discriminate healthy individuals and patients with cognitive disorders (including aMCI and AD) and non-AD (including controls and aMCI), and AD with high specificity and high sensitivity. The authors used the system to monitor the disease progression of six aMCI patients for over four years as well. Three of them persisted in the aMCI state, while the other three converted into AD. The analysis via the SPRi system of the patients’ plasma samples indicated that signals from the first three individuals were comparable to the ones evaluated earlier, while the signals from the AD patients decreased significantly respect with the samples collected before the disease transition. All these results demonstrated that the SPRi signal decrease can be used as a valid tool for the dynamic monitoring of these pathologies.

### 3.2. Parkinson Disease

In 2016, Yazdani and colleagues [48] designed three peptoid libraries, synthesized using a split and pool method, aiming at identifying antibody biomarkers in the blood of PD patients. For this study, serum samples from 75 PD patients, 25 de novo PD patients, and 104 normal control subjects from the NINDS Parkinson’s Disease Biomarker were examined. The peptoid PD2 (Table 2) from library 2 was identified as an encouraging candidate which is able to bind significantly higher levels of IgG3 antibodies in PD versus healthy subjects with a 68% accuracy; this is improved up to 84% in the identification of de novo PD. Later, the authors tried to validate the results by using a larger sample of individuals (99 de novo PD and 99 NC) followed longitudinally in the Parkinson’s Progressive Marker Initiative (PPMI) [49]. In the last case, the authors found no significant difference in levels of antibodies caught by the PD2 peptoid in the de novo PD with respect to control subjects and almost the same IgG3 serum levels in the two groups. Consequently, even if a correlation between IgG3 levels and PD2 peptoid-binding antibody levels was proved, PD2 peptoid cannot discriminate PDs from NCs.

In 2019, Gao and coworkers identified the peptoid ASBP-7 (α-syn binding peptoid-7) (Table 2) from a combinatorial library, showing high-affinity and selectivity for α-synuclein (α-syn). This is a well-known hallmark of PD that under physiological conditions aggregates to form insoluble fibrils leading to neurotoxicity and loss of the dopaminergic neurons [50]. The peptoid sequence designed for the library was formed by a variable part at the N-terminus containing four randomized monomers, a constant portion at the C-terminus including N-Lys for improving the solubility, and Cys residue for allowing the selective and covalent attachment to the metal surface SPRi chip. The peptoid library was synthesized and the evaluation of the binding affinity for the α-syn by kinetic studies was performed. ASBP-7 was selected as the best candidate to act as a potential selective probe for detecting PD. Then the ability of the peptoid to discriminate the sera of PD patients from those of the age-matched normal individuals (25) was checked using SPRi. The binding specificity of ASBP-7 for α-synuclein was confirmed by testing the interaction with the most abundant serum proteins such as albumin, IgG, transferrin, and IgM and another isoform of synuclein as well. Also, the authors evaluated the sensitivity of the peptoid to identify PD sera by using diverse dilution ratios of PD and normal sera indicating that at a high dilution ratio, the peptoid is still able to distinguish PD individuals from the healthy ones, reinforcing the potential of the system as a label-free probe to diagnose PD.

### 3.3. Neuromyelitis Optica

Neuromyelitis optica is a rare autoimmune demyelinating disease that can cause paralysis and blindness. The finding that the majority of NMO patients have high-levels of circulating IgG autoantibodies against the water channel protein aquaporin 4 (AQP4) expressed on the surface of astrocytes in the central nervous system was a significant advancement in the knowledge of NMO [51]. Currently, different methods for detecting anti-AQP4 autoantibodies are available: flow cytometric assays, ELISA against recombinant AQP4 protein, AQP4-transfected cell-based assays, tissue-based immunofluorescence, and fluorescence immunoprecipitation assays [52,53,54,55]. Despite the great diagnostic specificity of these various assays, approximately 25% [55] of patients with clinical NMO [56] lack easily detectable anti-AQP4 antibodies. Raveendra et al. [57] identified a panel of peptoids that bind anti-AQP4 antibodies in the serum of NMO patients by using a chemical library screening on beads, an approach that does not limit the screen to a few thousand molecules like in the microarray technology. The library of peptoids used encompasses five variable positions following an invariant linker of four residues. Initially, the library was exposed to a pool of six serum samples obtained from control individuals who are not affected by NMO, to exclude the beads that can bind to the antibodies found at high levels in control (non-NMO) sera. After that, a pool of NMO serum with a high concentration of anti-AQP4 antibodies was used to screen the denuded library. Red quantum dot-conjugated secondary antibodies were used to visualize antibody-binding beads in both the pre-screen and the NMO screen and several hits were selected and identified by tandem mass spectrometry. Ten of the forty-three hits that matched the brightest beads during the screening stage were resynthesized and spotted on chemically modified glass slides and then incubated with individual serum samples and, after washing, fluorescently labeled secondary antibodies. When the array was exposed to serum from healthy individuals there was little signal observed on any of the arrayed peptoids. The same experiment, performed on serum derived from NMO patients, who both tested positive and negative for anti-AQP4 antibodies via the cell-killing assay, allowed for the selection of four peptoids NMOP2, NMOP5, NMOP6, and NMOP9 (Table 2). Since the anti-AQP4 antibodies panel in patients is polyclonal and small molecules like peptoids are probably able to recognize only a portion of this polyclonal population, it is possible that, in larger studies, the use of a four-peptoids panel might be advantageous to bind different antibodies. Moreover, the ability of the panel to discriminate between NMO patient serum and serum from healthy controls or patients with narcolepsy, MS, AD, and lupus Erythematosus was also demonstrated.

### 3.4. Systemic Lupus Erythematosus

The diagnosis of autoimmune diseases like systemic lupus erythematosus is very challenging. The symptoms of the patients are not specific, and the clinically accessible autoantibodies lack the sensitivity or specificity to be utilized as diagnostic tests in the absence of disease-defining clinical characteristics. Almost all SLE patients can be identified by the presence of anti-nuclear antibodies (ANA), but they are not specific and can also be present in other autoimmune diseases and at low titer in healthy adults. Other autoantibodies are more specific for SLE but they are only found in a fraction of lupus patients on presentation [58,59]. Nevertheless, it is known that ANA and other autoantibodies appear years before the onset of clinical SLE and potentially forecast the shift from unaffected subjects to those with the pathology [60,61]. Unfortunately, a lack of specificity hampers their use as valid prognostic biomarkers. Quan and coworkers identified [62] several peptoids to identify individuals with lupus compared to controls with high sensitivity and specificity and with test properties similar to or better than traditional serological tests. Two different peptoid libraries consisting of 10 mer compounds were synthesized using a ‘split and pool’ strategy to catch antibodies that can characterize patients with SLE without considering the antigenic specificity of the antibody, and were screened via the magnetic screening method [63] opportunely modified. The first library was initially depleted of beads that display compounds able to bind to immunoglobulin from a pool of 46 ANA-negative healthy individuals and then incubated with the serum from 47 patients affected by SLE. The second library was depleted of peptoids that bound to a pool of 18 serum samples from healthy individuals with high levels of ANA to eliminate ligands that would interact with these autoantibodies but would not characterize diseased individuals. Then the library was incubated with a pool of serum from 19 patients with incomplete lupus erythematosus (ILE) showing the earliest signs of systemic autoimmunity. SLE15, ILE2, and ILE7 peptoids (Table 2) were selected. Validation of SLE15 and ILE2 was performed via the ELISA assay for their ability to bind IgG. The results indicate that the two classes of peptoids, SLE15 and ILE2/7, are capable of detecting various immunoglobulin biomarkers that can differentiate between various autoimmune diseases, such as early from developed SLE with moderate sensitivity and excellent specificity. The IgG bound to the SLE-peptoid was found to react with several autoantigens, indicating that the peptoids can interact with many molecules that share structural similarities.

### 3.5. Autism Spectrum Disorder

Autism spectrum disorder is a neurodevelopmental disorder associated with limitations in social interaction and communication, and restricted, repetitive behavior patterns (American Psychiatric Association. *Desk Reference to the Diagnostic Criteria from DSM-5.* 5th ed. Washington, D.C., American Psychiatric Association (2013)). Autism affects children aged 1 to 4 years and early diagnosis is required to reduce the burden of the pathology for the patients and their families [64,65]. Unlike other neurological diseases, biomarkers search for ASD is still recent. In 2016, Zaman et al. [66] investigated candidate antibody biomarkers related to ASD by exploiting previous approaches applied for the identification of NMO, SLE, and AD diseases [40,43,57,62]. They synthesized three different one-bead one-compound peptoid libraries screened by using a modification of the magnetic capture method [63].

The first library made up of 10-mer molecules containing seven changeable peptoid residues to ensure diversity was first depleted of peptoids that bound IgG in serum pooled from 10 typically developing (TD) male children. These were then incubated with serum pooled from 10 ASD males. Compounds able to bind IgG were considered “hit” peptoids. To reduce the number of the nonspecific “hit” beads produced during screens, a second library was designed to show less hydrophobic properties by the inclusion of the charged residue, NLys. Lastly, a third library was synthesized with a chemical diversity lower than the previous two to allow the isolation of redundant compounds during screening as an intermediate check for verifying the specificity of “hits”. ELISA assay was performed to assess the ability of the “hit” peptoids to bind IgG from the two pools of serum. Unexpectedly, the results obtained showed that the 25 peptoids isolated from the different screens can bind higher levels of IgG from the TD pool than from the ASD pool, independently from the library used. The peptoid ASD1 included in the first library (ASDn) (Table 2) was selected to further evaluate the nature of this differentiation. Then the authors looked at the binding of ASD1 to a small sample of females and the results indicated that peptoid levels were similar in the TD females and the TD males. There was no significant decrease in ASD1 binding in the ASD female versus TD subjects (both males and females). Unfortunately, the main limitation correlated to this study is due to the higher prevalence of ASD in men; thus, making it unable to fully examine any gender-specific variations. However, the results indicate that gender-specific differences seem to exist even with the very small sample size, but this observation must be confirmed. Further studies are required to evaluate whether the ASD1 peptoid, in combination with inflammatory blood analytes, can efficiently identify ASD.

**Table 2 ijms-24-16333-t002:** Peptoid sequences for blood-based screening. The structure of each monomer is shown in Figure 5 and the respective amines that were used to synthesize them are shown in Table 3.

Name	Sequence	Ref.
ADP1	*N*Leu-*N*Ser-*N*Lys-*N*Leu-*N*spe-*N*spe-*N*Lys-*N*Lys-Cys-NH_2_	[40]
ADP2	*N*all-*N*Ser-*N*pip-*N*all-*N*all-*N*all-*N*Lys-*N*Lys-Cys-NH_2_	
ADP3	*N*Lys-*N*pip-*N*Leu-*N*Lys-*N*Lys-*N*spe-*N*Lys-*N*Asp-Cys-NH_2_	[40,43]
Self-assembled Peptoid nanosheet with ADP3 loops	(…) *-*N*Lys-*N*pip-*N*Leu-*N*Lys-*N*Lys-*N*spe-*N*Lys-*N*Asp-*N*Glu-(…) *	[46,47]
PD2	*N*Lys-*N*Ser-*N*Lys-*N*Leu-*N*pyr-*N*mea-*N*dmpa-*N*mea-Cys-NH_2_	[49]
ASBP-7	*N*Lys-*N*a-*N*Glu-*N*pip-*N*Lys-*N*Lys-Cys-NH_2_	[50]
NMOP2	*N*api-*N*mba-*N*Lys-*N*pip-*N*bsa-*N*Lys-*N*Lys-*N*fur-Met-Cys-NH_2_	[57]
NMOP5	*N*api-*N*mba-*N*bsa-*N*pip-*N*bsa-*N*Lys-*N*Lys-*N*fur-Met-Cys-NH_2_	
NMOP6	*N*Lys-*N*api-*N*pip-*N*pip-*N*pip-*N*Lys-*N*Lys-*N*fur-Met-Cys-NH_2_	
NMOP9	*N*bsa-*N*Lys-*N*pip-*N*Lys-*N*mba-*N*Lys-*N*Lys-*N*fur-Met-Cys-NH_2_	
SLE15	*N*mba-*N*Lys-*N*dmpa-*N*pm-*N*cha-*N*Lys-*N*pip-*N*Lys-*N*Lys-Cys-NH_2_	[62]
ILE2	*N*pm-*N*fur-*N*pip-*N*mpa-*N*Leu-*N*Leu-*N*mpa-*N*mea-*N*mea-Cys-NH_2_	
ILE7	*N*pip-*N*fur-*N*pm-*N*all-*N*Ser-*N*all-*N*fur-*N*mea-*N*mea-Cys-NH_2_	
ASDn	(NR1-NR3-NR3-NR4-NR3-NR2-NR1) **-Nmea-Nmea-Met	[66]

* The ADP3 peptoid was embedded into the middle of the amphiphilic peptoid strand, named as (…), that was made up of the N-(2-carboxyethyl) glycine and N-(2-aminoethyl) glycine as the hydrophilic monomers and the N-(2-phenylethyl) glycine and N-(2-(4-biphenyl)ethyl) glycine as the hydrophobic monomers. ** where R_n_ are: Nall, NAsp, Ncha, Ndmpa, Nffa, NLeu, Nmba, Nmpa, Npip, Npyr, NSer.

**Table 3 ijms-24-16333-t003:** List of monomer acronyms and their corresponding amine precursors.

Monomer Acronym	Amine Precursor
**Na**	1-naphthylamine
**Nall**	allylamine
**Napi**	1H-imidazole-4-propylamine
**NAsp**	Glycine
**Nbsa**	4-(2-aminoethyl)benzenesulfonamide
**Ncha**	cyclohexylamine
**Ndmpa**	3,4-dimethoxyphenethylamine
**Nfur = Nffa**	furfurylamine
**NGlu**	b-alanine
**NhomoTyr**	p-hydroxyphenylethylamine
**NLeu**	isobutylamine
**NLys**	1,4-diaminobutane
**Nmea**	2-methoxyethylamine
**Nmpa**	3-methoxypropylamine
**Npip**	piperonylamine
**Npm**	benzylamine
**Npyr**	N-(3′-aminopropyl)-2-pyrrolidinone
**Nser**	ethanolamine
**Nspe = Nmba**	(R)-methylbenzylamine

## 4. Conclusions

In this review, we extensively described peptoid-based approaches for the diagnosis of cancers, neurological and autoimmune disorders. Early detection of biomarkers associated with these diseases is of utmost importance for the development and implementation of effective treatments, which can relieve the progression of pathology and enhance the patient’s quality of life. A biomarker is employed for the comprehension of crucial and key biological processes and their relationship with disease patho-physiologies and therapeutic regimens. As new tools for creating diagnostic and therapeutic regimens, specific biomarkers are detected by different targeted molecular imaging techniques, and by screening of plasma, serum, and tissue samples. The development of adequate molecular probes to detect disease-related biomarkers throughout the body is still challenging. In the search of new eligible molecular probes, peptoids represent appealing candidates to focus on for obtaining smart tools with high-specificity, selectivity, stability, and versatility, all included in a single molecule class. Here we broadly reviewed the most representative peptoids with high-potential as probes for diagnosis of cancers, neurological and autoimmune syndromes, showing that the results reported in the collected literature strongly encourage future investigations.

## Figures and Tables

**Figure 1 ijms-24-16333-f001:**
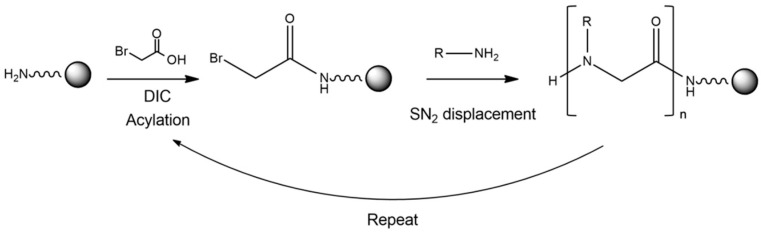
Schematic representation of the synthetic strategy used to obtain a peptoid. In particular, a primary amine on the resin is acylated via an activated haloacetic acid, such as bromoacetic acid, with N,N-diisopropylcarbodiimide (DIC). Then, the bromine is displaced via a primary amine during a SN2 reaction.

**Figure 2 ijms-24-16333-f002:**
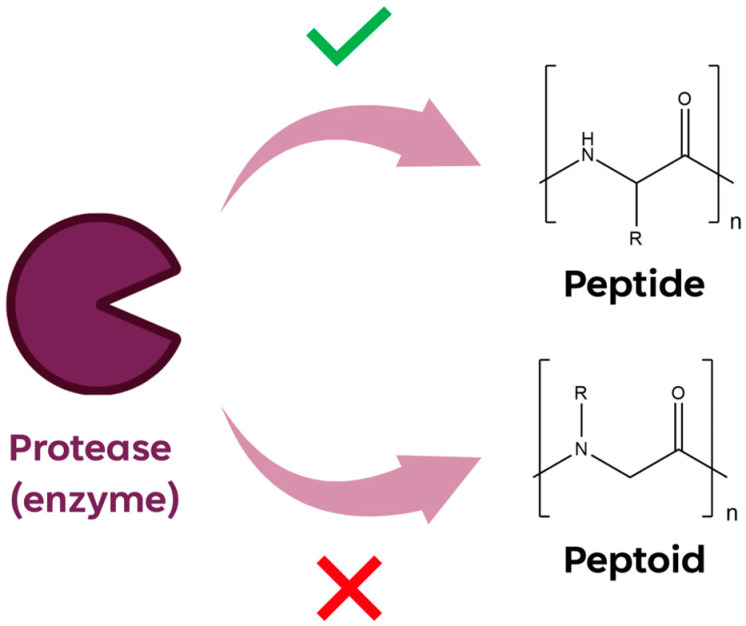
Graphic representation of enzymatic proteases cleavage on peptide and peptoid sequences.

**Figure 3 ijms-24-16333-f003:**
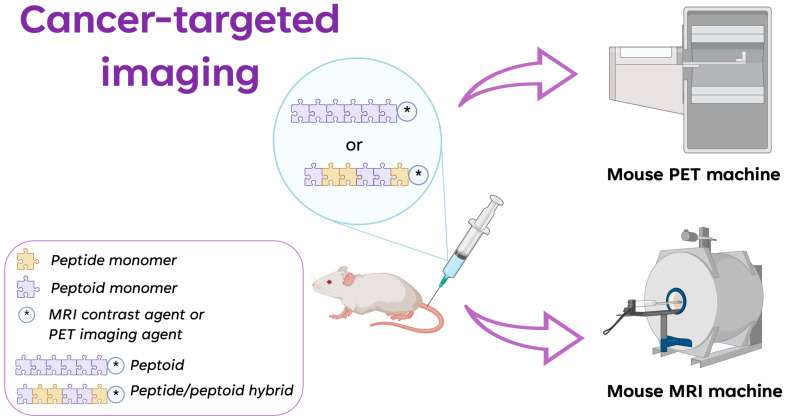
Graphic representation of the application of peptoids and peptide/peptoid hybrids in cancer diagnosis by using PET and MRI as imaging techniques. * is the contrast agent in MRI and the imaging agent in PET (created with Biorender.Com).

**Figure 4 ijms-24-16333-f004:**
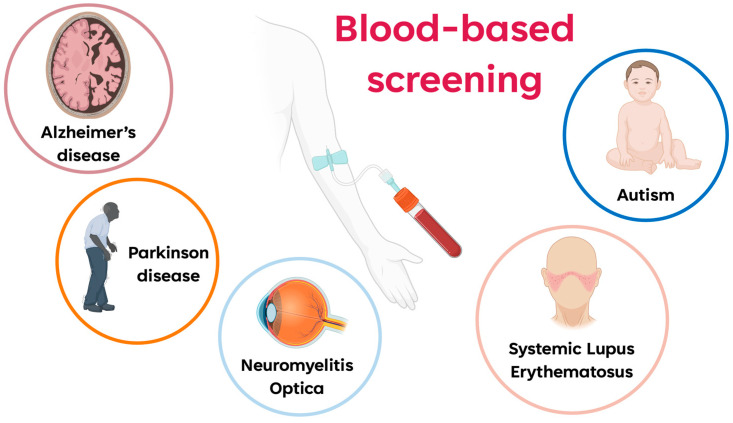
Graphic representation of different pathologies being studied through a blood-based screening using peptoid oligomers (created with BioRender.com).

**Figure 5 ijms-24-16333-f005:**
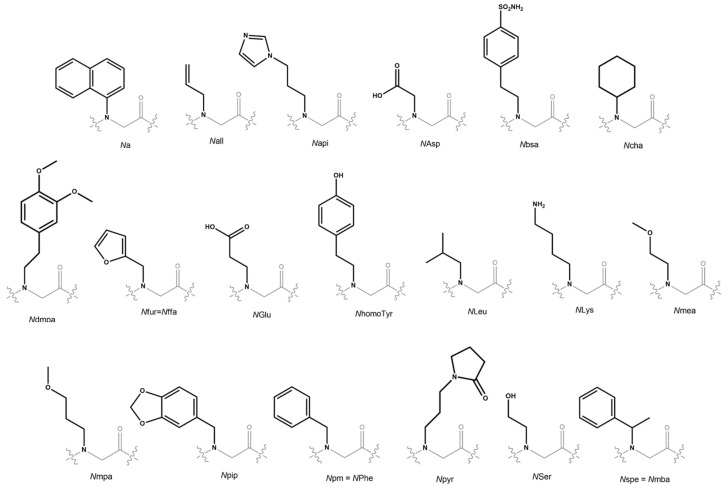
Structures of all peptoid monomers cited in this review.

## Data Availability

No new data were created or analyzed in this study. Data sharing does not apply to this article.

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
