# Peer review of "Peptoids: Smart and Emerging Candidates for the Diagnosis of Cancer, Neurological and Autoimmune Disorders"

_ijms, 2023, doi:10.3390/ijms242216333_

Round 1
Reviewer 1 Report
Comments and Suggestions for Authors
Known in the field based on previous literatures:
1. Peptoids or N-substituted glycines, a class of peptidomimetic molecules, are versatile tools to probe biological processes and hold potential as therapeutic agents.
2. Biological markers, also called biomarkers, are quantitative measurements that provide information about a disease state. These markers provide much-needed insight into preclinical and clinical data and vital for diagnostics of brain diseases.
3. Early detection and accurate diagnosis are essential for the early intervention of any diseases, particularly neurological disorders. Hence, novel blood biomarkers help in the understanding of neurological disorders and to identify at initial stages before symptoms.
In this review article authors discuss following findings:
I have gone through the review titled ‘Peptoids: smart and emerging candidates for the diagnosis of cancer, neurological and autoimmune disorders’. Authors discuss the recent advancements regarding the potential applications of peptoid as a useful platform for cancer targeted imaging by PET, MRI and blood-based screening in the diagnosis of neurological disorders. Authors discussed following main points-
1. synthetic strategy used to get a peptoid and their application in diagnosis.
2. discussed about radiolabeled peptoid and peptoid hybrid sequences used for cancer-targeted imaging and different peptoids used for blood-based screening of different neurological and autoimmune diseases.
Although, there is nothing new but the facts and material presented are interesting and generally supportive of the conclusions drawn. However, some issues that require the authors' attention. The following suggestions if incorporated could help in the better understanding of the significance of the work and implications.
Minor/major Concerns:
1. Authors should rephrase the sentence (line 95-96)- Most neurological dysfunction occurs usually by imaging, blood markers, and cerebrospinal fluid detection. They should write- most neurological dysfunction exam and test occur by imaging, blood markers, and cerebrospinal fluid detection.
2. Authors should briefly cover all the peptoides studied before in neurological diseases. As for example, they should discuss about PD2 peptoid, useful for Parkinson’s disease early-stage identification and serve as an indicator of disease severity.
3. Authors should clearly mention about how this review different from rest? Does it embrace a specific gap in the field as compared to previous review?
Author Response
Minor/major Concerns:
- Authors should rephrase the sentence (line 95-96)- Most neurological dysfunction occurs usually by imaging, blood markers, and cerebrospinal fluid detection. They should write- most neurological dysfunction exam and test occur by imaging, blood markers, and cerebrospinal fluid detection.
- Authors should briefly cover all the peptoids studied before in neurological diseases. As for example, they should discuss about PD2 peptoid, useful for Parkinson’s disease early-stage identification and serve as an indicator of disease severity.
- Authors should clearly mention about how this review different from rest? Does it embrace a specific gap in the field as compared to previous review?
Authors' Reply to the Reviewer
Thanks to the reviewer for giving us the opportunity to improve the manuscript.
The authors generated a track changes version of the manuscript. The bibliography’s changes have not been evidenced as track changes.
1) The sentence Line 95-96 has been rephrased.
2) The authors have discussed about PD2 peptoid reported as a further candidate for Parkinson’s disease early-stage identification.
3) In the introduction section the authors have highlighted that the present review is the only one reported in the last decade in the field of peptoids in the diagnosis.
Reviewer 2 Report
Comments and Suggestions for Authors
This manuscript has comprehensively shown the employment of peptoids in targeted cancer imaging and blood-based screening of neurological and autoimmune diseases. This topic is of great importance to the life science and chemistry community, so I think this paper can be accepted after some following minor issues:
1. Some of the cartoon fonts in the figure are hard to read and some letters are too small. The authors should adjust it properly.
2. Some outlook of this field should be provided.
Author Response
This manuscript has comprehensively shown the employment of peptoids in targeted cancer imaging and blood-based screening of neurological and autoimmune diseases. This topic is of great importance to the life science and chemistry community, so I think this paper can be accepted after some following minor issues:
- Some of the cartoon fonts in the figure are hard to read and some letters are too small. The authors should adjust it properly.
- Some outlook of this field should be provided
Authors’ Reply to the Reviewer
The authors generated a track changes version of the manuscript. The bibliography’s changes have not been evidenced as track changes.
The authors thank the reviewer for the suggestions
- The font and the size have be changed in the cartoons (figure 2 and Figure 3)
- In the introduction section the authors have highlighted that the present review is the only one reported in the last decade in the field of peptoids in the diagnosis
Reviewer 3 Report
Comments and Suggestions for Authors
This review describes the recent development of peptoids as tracers for the diagnosis of multiple diseases. The topic is very novel and interesting, and the writing is well structures. I have some additional suggestions to further improve the quality of the manuscript.
1. In the introduction section, one page 3, there are two paragraphs that talk about MRI and neurological/autoimmune diseases, but they do not have any references. For every piece of information provided in the manuscript, as long as it is not a common sense, there should be a reference to back up the statement.
2. Figure 3 and Figure 4 are referred to at the end of the introduction section, whereas a more reasonable way is to refer to Figure 3 in section 2 and Figure 4 in section 3.
Author Response
This review describes the recent development of peptoids as tracers for the diagnosis of multiple diseases. The topic is very novel and interesting, and the writing is well structures. I have some additional suggestions to further improve the quality of the manuscript.
- In the introduction section, one page 3, there are two paragraphs that talk about MRI and neurological/autoimmune diseases, but they do not have any references. For every piece of information provided in the manuscript, as long as it is not a common sense, there should be a reference to back up the statement.
- Figure 3 and Figure 4 are referred to at the end of the introduction section, whereas a more reasonable way is to refer to Figure 3 in section 2 and Figure 4 in section 3.
Authors’ Reply to the Reviewer
The authors generated a track changes version of the manuscript. The bibliography’s changes have not been evidenced as track changes.
- In the introduction section in page 3 one reference about MRI (reference 18) and two references about neurological/autoimmune diseases (references 19, 20) have been inserted.
- Figure 3 and figure 4 have been moved to section 2 and 3, respectively